# Plasmodium vivax transcriptomes reveal stage-specific chloroquine response and differential regulation of male and female gametocytes

Adam Kim[1], Jean Popovici [2], Didier Menard [2,3] & David Serre [1]

Studies of Plasmodium vivax gene expression are complicated by the lack of in vitro culture system and the difficulties associated with studying clinical infections that often contain multiple clones and a mixture of parasite stages. Here, we characterize the transcriptomes of P. vivax parasites from 26 malaria patients. We show that most parasite mRNAs derive from trophozoites and that the asynchronicity of P. vivax infections is therefore unlikely to confound gene expression studies. Analyses of gametocyte genes reveal two distinct clusters of co-regulated genes, suggesting that male and female gametocytes are independently regulated. Finally, we analyze gene expression changes induced by chloroquine and show that this antimalarial drug efficiently eliminates most P. vivax parasite stages but, in contrast to P. falciparum, does not affect trophozoites.

[1] Institute for Genome Sciences, University of Maryland School of Medicine, 670 W Baltimore Street, Baltimore, MD 21201, USA. [2] Malaria Molecular Epidemiology Unit, Institut Pasteur in Cambodia, 5 Boulevard Monivong, PO Box 983, Phnom Penh 12 201, Cambodia. [3] Present address: Biology of Host-Parasite Interactions Unit, Institut Pasteur, 25-28 Rue du Dr. Roux, 75724 Paris, France. Correspondence and requests for materials should be addressed to D.S. (email: dserre@som.umaryland.edu)

*P*lasmodium vivax is the most widespread human malaria parasite, responsible for more than 8.5 million clinical malaria cases worldwide in 2016 and threatening more than 2 billion people in 90 countries[1,2]. *P. vivax* diverged more than 100 million years ago from *Plasmodium falciparum*[3] and displays unique biological features complicating malaria treatment and elimination, including a lower parasitemia and pyrogenic threshold[4], an elusive dormant liver-stage[5], and a rapid gametocytogenesis[6] often leading to the apparition of transmittable parasites before the onset of malaria symptoms. Unfortunately, our understanding of the biology of this important human pathogen remains limited and lags behind that of *P. falciparum*, primarily due to our inability to continuously propagate *P. vivax* parasites in vitro[4]. Most studies of *P. vivax* have to rely on infected blood samples, which complicates biological and molecular investigations due to the polyclonality of many *P. vivax* infections[7–14], the concurrent presence of different parasite stages (caused by the lack of sequestration[4,15] typical in *P. falciparum*[16]) with their specific regulatory programs and responses, and the abundance of host molecules that hamper genomic studies. As a result, studies of *P. vivax* transcriptomes, which could provide unique insights on the biology of this parasite and its differences with *P. falciparum* and rodent parasites, have been few and far between, and limited to parasites grown in short-term ex vivo culture[17–19] (which might alter the gene expression profiles[20]). We therefore still have a very limited understanding of the patterns of *P. vivax* gene expression during clinical infections or of their changes upon antimalarial drug treatment.

We previously analyzed blood samples from three *P. vivax* infected patients and showed that, despite extensive differences in stage composition, the parasite gene expression profiles were very similar across infections[21]. Here, we expand these analyses and characterize the in vivo parasite gene expression profiles from 26 vivax malaria patients enrolled in a chloroquine efficacy study[22,23]. First, we use gene expression deconvolution to estimate the stage of origin of the transcripts characterized and rigorously address composition heterogeneity. Second, we analyze variations in the expression of gametocyte genes among infected patients to better understand the mechanisms underlying successful parasite transmission. Finally, we compare the gene expression profiles of parasites before and after chloroquine administration to gain insights on the drug mode of action and examine how *P. vivax* parasites respond to this therapeutic stress. Overall, our study provides a global perspective on the diversity of expression profiles of *P. vivax* parasites in vivo and of their regulation.

## Results

**Variations in *P. vivax* gene expression among infections**. We extracted RNA from ~50 μL of whole blood collected from 26 Cambodian individuals seeking treatment for vivax malaria. All patients were tested positive for *P. vivax* by RDT and blood smears, and PCR confirmed that they were solely infected with *P. vivax*[22,23]. We prepared and sequenced a stranded RNA-seq library from each blood sample after globin and rRNA reduction[21]. After removing reads originating from human transcripts, we aligned all remaining reads to the most recent *P. vivax* genome sequence[24]. The percentage of reads originating from *P. vivax* transcripts varied greatly among samples, from 1.09% to 41.63% (median: 11.97%), with only a moderate correlation with the samples' parasitemia (Pearson's $R = 0.37$, $p = 0.06$, Supplementary Figure 1). After stringent quality filters, 253,914–36,491,854 reads aligned to the *P. vivax* genome and 20 of the 26 samples yielded more than one million *P. vivax* reads (Supplementary

Data 1). Out of the 6823 annotated *P. vivax* genes, 4999 were deemed expressed in at least ten patients and were further analyzed. Principal component analysis of the gene expression profiles showed no clear separation between samples (Supplementary Figure 2), nor according to the parasitemia, gametocytemia, or the stage composition determined by microscopy.

*P. vivax* infections are typically asynchronous and, at enrollment, most patient blood samples displayed multiple parasite stages in variable proportions (Fig. 1a). To statistically determine the contribution of the different developmental stages present in each infection to the overall *P. vivax* expression patterns, we performed gene expression deconvolution using single-cell RNA-seq data from *P. falciparum* synchronized cultures[25] (see Methods). We first used Cibersort[26] to determine the signature gene expression profile of each developmental stage using reference datasets (in our case from single-cell *P. falciparum* transcriptome data) and used these signatures to infer the relative contributions of these different cell types to bulk RNA-seq data. To validate this approach, we first analyzed RNA-seq data from synchronized *P. falciparum* cultures[27] as well as artificial bulk RNA-seq data generated by mixing, in silico, known proportions of single-cell RNA-seq from *Plasmodium berghei*[28]. Overall, the deconvolution results from these control datasets were concordant with the expected composition of these samples (Supplementary Figure 3), indicating that gene expression deconvolution analysis is able to infer the stage of origin of the transcripts characterized from an asynchronous infection. We then used this approach to determine the stage composition of each *P. vivax* clinical infection based on its parasite gene expression profile. Consistent with our previous analyses[21], we observed that, in each patient infection, the parasite transcripts derived almost exclusively from trophozoites, regardless of the stage composition determined by microscopy (Fig. 1b).

**Regulation of the expression of gametocyte genes**. The reference gene expression datasets used for gene expression deconvolution did not include gametocytes and the contribution of sexual parasites to the overall profile of each sample could therefore not be estimated using this approach. Examination of the transcript levels of the gametocyte genes Pvs47 and Pvs48/45[29] revealed highly correlated expression among patients (Fig. 2a, Pearson's $R = 0.93$, $p < 0.01$). To systematically investigate this pattern, we first compared the gene expression levels of 21 *P. vivax* genes thought to be expressed in gametocytes (Table 1). The gene expression of these 21 predicted gametocyte genes was highly correlated among samples but, interestingly, clustered into two distinct subsets poorly correlated with each other (Fig. 2b). Thus, Pvs47, Pvs48/45, Hap2, the gamete egress and sporozoite traversal protein, s16, and three CPW-WPC proteins were all positively correlated with each other (Cluster A in Fig. 2b, Pearson's $R = 0.15–0.96$, $p < 0.01$), while Pvs25, ULG8, the gametocyte associated protein, gametocyte developmental protein 1, guanylate kinase, HMGB1, and five CPW-WPC proteins all clustered into a second group (Cluster B, Pearson's $R = 0.05–0.86$, $p < 0.01$). Only genes belonging to this second group (Cluster B) had their expression correlated with the gametocytemia determined by microscopy (Pearson's $R > 0.5$, $p < 0.01$, Table 1). To expand our knowledge of *P. vivax* genes possibly expressed in gametocytes, we searched for additional genes whose expression correlated with any of these gametocyte genes. Overall, the expression of 1613 genes was correlated with that of at least one of these known gametocyte genes (Pearson's $R > 0.8$, $p < 0.01$) and could represent putative novel gametocyte genes. These included 460 genes correlated with members of Cluster A and 1153 correlated with genes in Cluster B (Supplementary Data 2). Gene ontology

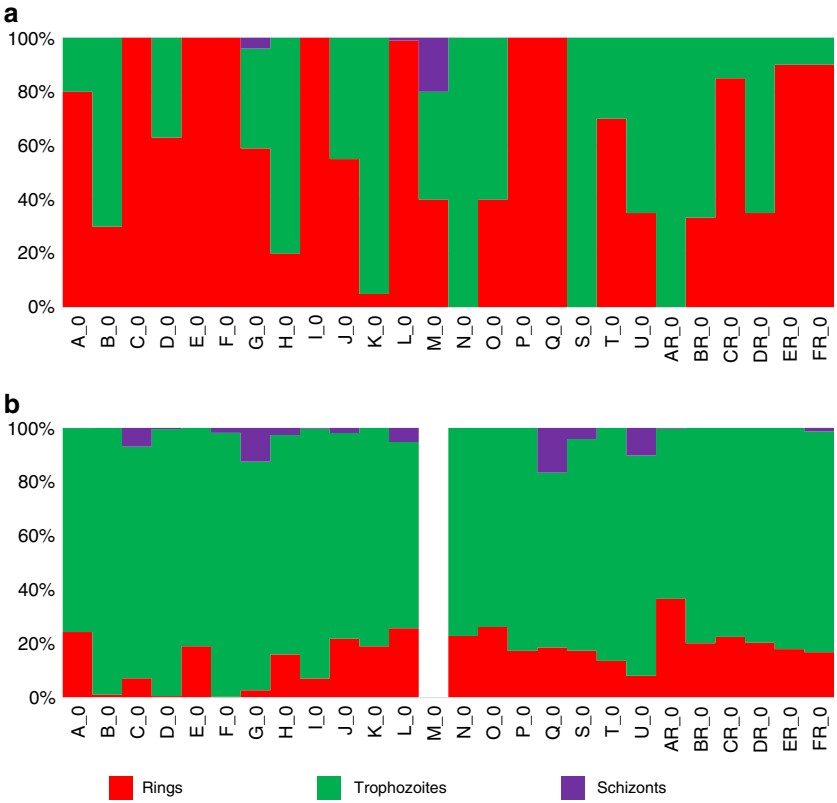

**Fig. 1** *P. vivax* stage composition of all clinical infections. Each vertical bar represents one clinical infection and is colored according to the proportion of schizonts (purple), trophozoites (green and blue), and rings (red). The top panel displays the stage composition determined by microscopy (**a**) while the bottom panel shows the stage composition inferred from gene expression deconvolution of the same infections (**b**). (The gene expression deconvolution for the M infection was not successful and is not shown.)

analyses showed that genes whose expression correlated with those of gametocyte genes from Cluster A were enriched in biological processes such as microtubule-related genes, including dynein, kinesin, and tubulin. By contrast, genes associated with intracellular trafficking and histone remodeling were over-represented in Cluster B.

**Effect of chloroquine exposure on *P. vivax* transcriptome**. To assess how *P. vivax* parasites respond to chloroquine exposure, we compared the expression profiles of the parasites collected, from the same 20 infections, before and 8 h after the first dose of chloroquine. Consistent with the drug-induced clearance observed by microscopy, we observed that the proportion of RNA-seq reads aligned to the *P. vivax* genome decreased after chloroquine treatment (Fig. 3a). However, while the parasitemia measured by microscopy decreased by 57% (median) in the 8 h following the administration of the first dose of chloroquine, the proportion of *P. vivax* reads decreased by more than 81% (with no clear correlation with the change in parasitemia determined by microscopy, Supplementary Figure 4). Gene expression deconvolution analyses indicated that, 8 h after treatment, most parasite RNAs still primarily derived from trophozoites (Supplementary Figure 5).

To account for the extensive decrease in *P. vivax* reads post-treatment, that can lead to statistical artefacts, we randomly subsampled the datasets generated prior to chloroquine administration to the same number of reads as in the post-treatment datasets (see Methods for details). Principal component analysis showed that the parasite RNA profiles did not separate samples pre- and post-treatment (Fig. 3b), suggesting that, despite the large decrease in total parasite RNA induced by chloroquine, the

overall gene expression patterns were not dramatically affected qualitatively. Indeed, permutation analyses showed that the differences in gene expression between paired samples (i.e., before and after chloroquine from the same patient) were significantly lower than those between randomly paired samples (i.e., comparing before and after chloroquine samples from different patients) ($p = 4.2 \times 10^{-4}$). This result indicated that the combination of parasite genetic diversity and host response to the infection had a greater effect on the parasite gene expression than the exposure to chloroquine.

While it did not separate the samples collected before and after chloroquine, PC1 was statistically associated with chloroquine treatment in paired analyses ($p = 0.002$, Supplementary Figure 6) suggesting that chloroquine did influence parasite gene expression. We therefore tested the effect of chloroquine exposure on each annotated *P. vivax* gene. Out of 4280 genes tested, 293 genes (195 downregulated and 98 upregulated) showed a statistically significant change in expression following chloroquine treatment (FDR < 0.1, Fig. 3c). Interestingly, a large number of exported protein genes (including 23 PHIST genes) and PIR genes were significantly downregulated after treatment, as were many genes involved in erythrocyte invasion (e.g., AMA1, DBP, MSP5, RBP2a, RBP2e, or RBP3) (Supplementary Data 3). Genes upregulated included ribosomal RNAs and histones. None of the candidate genes suspected to be associated with chloroquine susceptibility in *P. vivax* or *P. falciparum* showed significant changes in expression: for example, the chloroquine resistance transporter gene (CRT) showed only a modest, non-significant decrease in expression (log2 fold change = −0.47, FDR = 0.42) as did the multidrug resistance gene (MDR1) (log2 fold change = −1.48, FDR = 0.27). (Note that previous clinical and genetic analyses suggested that

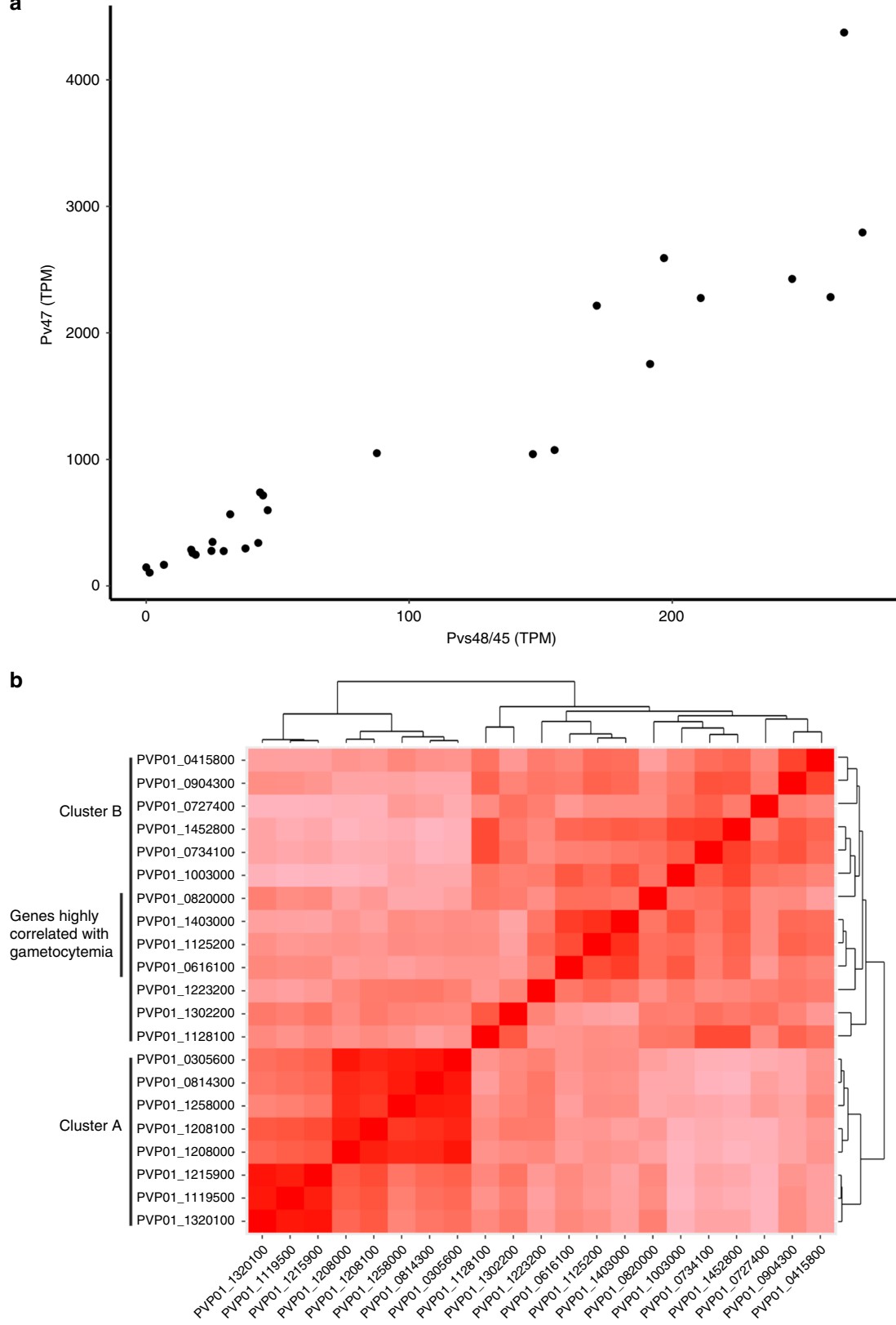

**Fig. 2** Expression of gametocyte genes. **a** Correlation between the expression level of two gametocyte genes. Each dot on the figure represents a single *P. vivax* patient infection and is displayed according to the normalized gene expression values of Pvs48/45 (*x*-axis) and Pvs47 (*y*-axis). **b** Heatmap showing the extent of gene expression correlation (Pearson's *R,* in red scale) among 21 gametocyte candidate genes. The bordering tree shows the results of unsupervised clustering of these genes according to their gene expression pattern. Genes whose expression levels are highly correlated with the gametocytemia determined by microscopy are also indicated

**Table 1 Clustering of known gametocyte genes**

| Cluster A | | Cluster B | |
| --- | --- | --- | --- |
| PVP01_0814300 | Male gamete fusion factor HAP2 | PVP01_0616100 | Ookinete surface protein Pvs25[a] |
| PVP01_0305600 | Sexual stage antigen s16 | PVP01_0820000 | CPW-WPC family protein[a] |
| PVP01_1258000 | Gamete egress sporozoite traversal protein | PVP01_1403000 | Gametocyte associated protein[a] |
| PVP01_1208100 | 6-Cysteine protein | PVP01_1125200 | CPW-WPC family protein[a] |
| PVP01_1208000 | 6-Cysteine protein | PVP01_0415800 | 6-Cysteine protein |
| PVP01_1215900 | CPW-WPC family protein | PVP01_0904300 | CPW-WPC family protein |
| PVP01_1119500 | CPW-WPC family protein | PVP01_0727400 | Guanylate kinase |
| PVP01_1320100 | CPW-WPC family protein | PVP01_1452800 | Upregulated in late gametocytes ULG8 |
| | | PVP01_0734100 | Gametocyte development protein 1 |
| | | PVP01_1003000 | CPW-WPC family protein |
| | | PVP01_1223200 | CPW-WPC family protein |
| | | PVP01_1302200 | High mobility group protein B1 |
| | | PVP01_1128100 | Inner membrane complex protein 1j |

[a]Indicates genes whose expression levels are correlated with the gametocytemia measured by microscopy

none of these *P. vivax* isolates were resistant to chloroquine[23].) Since we previously showed that both CRT and MDR1 could be spliced into multiple isoforms in *P. vivax*[21], we also tested whether the transcription of different isoforms was associated with chloroquine susceptibility. In samples pre-treatment, we did not observe any association between the isoforms expressed and the subsequent response to chloroquine: retention of the CRT intron 9, that leads to a predicted early stop codon, was highly variable among infections (Supplementary Figure 7A) but was not associated with the decrease in parasitemia ($p = 0.58$) nor with the proportion of *P. vivax* reads post-treatment ($p = 0.95$). Similarly, the extent of splicing of the 3′ UTR of MDR1 varied greatly among infections (Supplementary Figure 7B) but was not associated with the response to chloroquine.

## Discussion

RNA-sequencing has provided invaluable insights on some of the fundamental characteristics of *Plasmodium* gene expression and, for example, highlighted the complexity of these parasites' transcriptomes, with extensive noncoding transcripts, long 5′ and 3′ untranslated regions, and, often, multiple isoforms per gene[17,21,30–32]. Additionally, gene expression studies have been extremely useful to identify transcriptional differences between parasite stages[18,19,27,33,34], in response to antimalarial drugs[35–38] or to culture conditions[20,39–41]. However, most of these studies have been conducted using rodent malaria parasites or in vitro cultures of *P. falciparum*, and our understanding of gene regulation in other human malaria parasites remain very incomplete (but see also ref. [34]). This is notably true for *P. vivax* for which the lack of in vitro culture system restricts studies to patient samples, with all the associated problems due to overwhelming host RNA and asynchronous parasites.

In this study, we showed that parasite gene expression profiles could be consistently generated from patients presenting with clinical vivax malaria. In contrast with the variable stage composition observed by microscopy among patient infections, gene expression deconvolution analyses revealed that the vast majority of RNA molecules in a *P. vivax* blood infection derived from a single developmental stage, the trophozoites. This observation is consistent with our previous analyses of three *P. vivax*-infected patients[21], and with nuclear run-on experiments[31,42], RNA polymerase II profiling[31] and single-cell RNA-seq[25] that showed that *Plasmodium* trophozoites are much more transcriptionally active than the other asexual parasite stages. In addition, the relatively low sensitivity of microscopy likely contributes to the discrepancy between the microscopy and gene expression

deconvolution data: given the low parasitemia typical of *P. vivax* infections, only a small number of parasites are typically observed by microscopy, which might miss lowly abundant stages (e.g., the schizonts in Fig. 1) and limit the accuracy. For example, in an infection with a parasitemia of 0.1%, analysis of 200 fields by microscopy only provides stage information based on ~50 observed parasites, while RNA-seq data generated from 50 μL of blood would summarize information from ~250,000 parasites.

The overwhelming transcriptional signal of trophozoite parasites circumvents the limitation of studying asynchronous infections and indicates that analyses of parasite RNA-seq profiles generated directly from infected patient blood are unlikely to be confounded by stage differences (and could be further controlled by using the gene expression deconvolution results as covariates in the analyses). However, these results also imply that differences in gene regulation occurring in the less transcriptionally-active stages (e.g., rings) will be more difficult to detect using whole blood RNA-seq since fewer transcripts from these stages will be represented in the dataset (though our analysis of gametocyte genes show that this might not be impossible). Thus, analysis of a mechanism specific to ring-stage parasites (e.g., dormancy) might be difficult using whole-blood RNA-seq and may require other approaches, such as profiling after short-term cultures or single-cell RNA-seq[25,28], to specifically investigate gene expression regulation at this stage.

We believe that the gene expression deconvolution analyses described here could be very useful in many *Plasmodium* gene expression studies to identify, and correct for (if necessary), the stages from which the mRNA molecules are derived. However, it is important to emphasize the limitations of these analyses that rely on a number of assumptions. First, the accuracy of the deconvolution depends on the reference datasets used for determining the signature of each stage (in our case scRNA-seq generated from synchronized *P. falciparum* cultures[25]). Using these profiles to deconvolute datasets from other synchronized *P. falciparum* cultures[27] revealed some stage heterogeneity suggesting that (i) stage deconvolution is not entirely accurate and/or (ii) that some of these cultures might not have been perfectly synchronized and contained different parasite stages (Supplementary Figure 3). In addition, we used here profiles from *P. falciparum* to analyze *P. vivax* infections which may introduce some errors if the transcriptional regulation of the different stages differs between *Plasmodium* species. However, the analysis of mock *P. berghei* mixtures suggests that species differences are unlikely to dramatically influence deconvolution results, probably because the signature profile of each stage is a composite of hundreds of genes expressed at this stage (Supplementary Data 4) and is therefore robust to limited species-specific gene expression

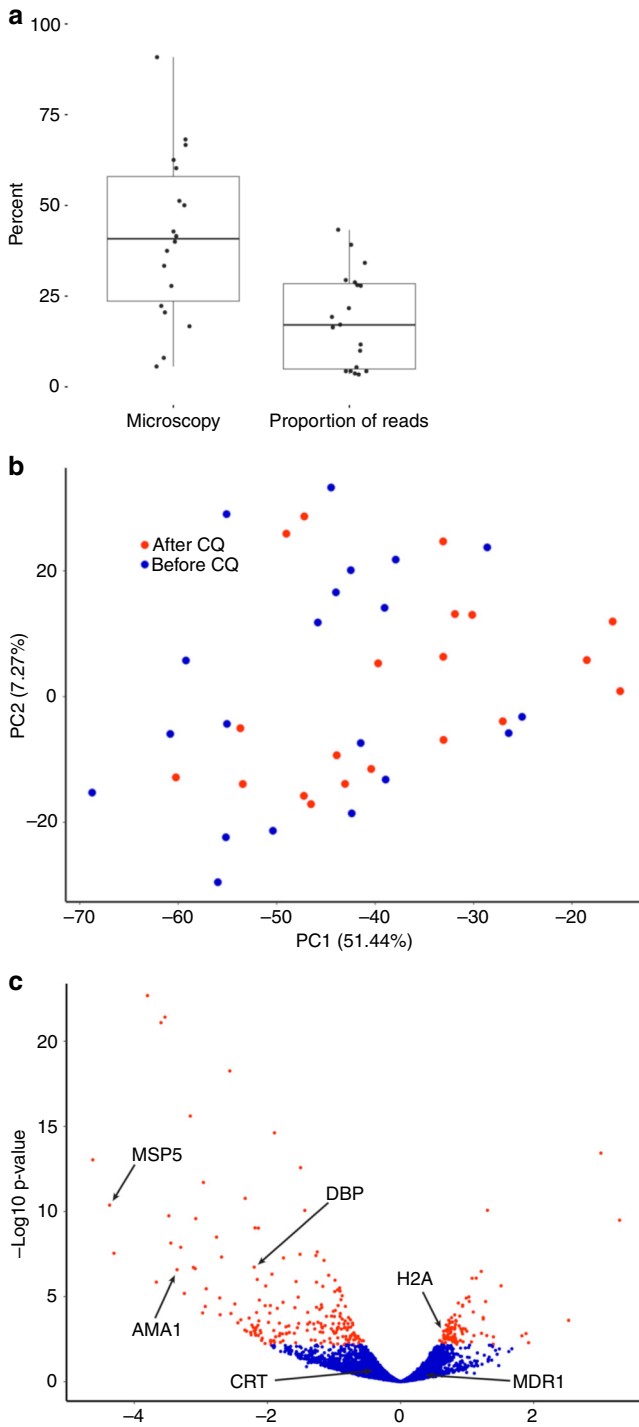

**Fig. 3** Effect of chloroquine on parasite gene expression. **a** Decrease in the number of parasites 8 h after chloroquine administration (represented by the proportion of remaining parasites, in %) measured by microscopy and by the proportion of parasite reads. Each dot represents one infection, the centerline shows the median, the box limits the quartiles, and the whiskers indicate the range (with outliers removed). **b** Principal component analysis of the parasite gene expression profiles from each sample before (blue dots) and 8 h after chloroquine administration (red dots). **c** Volcano plot of the gene expression changes after chloroquine treatment. Each dot represents one annotated *P. vivax* gene and is displayed according to the fold-change in expression (x-axis, in log2) and statistical significance (y-axis, in negative logarithm to the base 10 of the p-value). Red indicates significantly affected genes (FDR < 0.1)

differences[34]. Finally, it is possible that the reference profiles of in vitro parasites may not truly recapitulate in vivo gene expression patterns[20]. Note that all these potential issues could be alleviated in the near future by using, for the reference stage profiles, scRNA-seq data generated from *P. vivax* in vivo infections. We also used here gene expression deconvolution to determine the proportion of mRNA derived from each stage, assuming that these developmental stages are discrete entities, when in fact parasite gene expression varies continuously along the intraerythrocytic developmental stages[25,28,34,35,43] (but see also ref. [44]). However, we chose to classify parasites into discrete developmental categories to facilitate the comparison with microscopy data and the data interpretation (see e.g., the discussion of the effect of chloroquine below). Along the same idea, the discrete parasite stages used to categorize the origin of each transcript are crudely defined and likely encompass a range of more subtle developmental changes. This low resolution might fail to capture the exact biological mechanisms at play and, for example, the overall conservation of the gene expression profiles observed in broadly-defined trophozoites upon chloroquine exposure might mask differences among parasites at different points of this extended developmental period. Again, these limitations could be alleviated in future studies by grouping reference datasets from scRNA-seq into more bins, representing intermediary and transitional stages, and thus providing a better temporal resolution.

The dataset used for gene expression deconvolution did not include profiles from gametocytes and these were therefore not considered in these analyses. However, we observed that the expression levels of putative gametocyte markers were highly correlated with each other across infections but, surprisingly, clustered in two distinct groups. Our co-expression analyses also revealed novel putative gametocyte genes and indicated that microtubule-associated proteins were overrepresented among the genes from one cluster, while intracellular trafficking and histone remodeling genes were overrepresented in the second cluster. One explanation for these observations is that the Cluster A (that includes hap2) represented genes expressed in male gametocytes while Cluster B (that includes Pvs25) corresponded to female gametocyte genes (Table 1). This hypothesis is consistent with the observation that gametocytemia (determined by microscopy) was only correlated with the expression of genes from Cluster B since *P. vivax* female gametocytes are much more abundant and easier to detect by microscopy[45,46]. These data indicate that, not only do the proportions of gametocytes vary across infections but, more importantly, that the ratio of male to female gametocytes differs among infections. This observation is consistent with previous studies[47,48] and implies that the male and female terminal gametocytogeneses are independently-regulated, which could provide a mechanism for the parasites to reduce the probability of self-fertilization in mosquito by isolating, perhaps temporally, the generation of gametes of different sexes by a single clone. Note that by identifying genes that are potentially highly expressed in male and female *P. vivax* gametocytes (Supplementary Data 2), our findings will also facilitate further investigations of this hypothesis by enabling the monitoring, over time, of changes in the relative proportions of male and female gametocytes in *P. vivax* infections by qRT-PCR. In addition, these findings, if confirmed, would suggest that estimates of transmission potential need to account for both sexes, rather than just the presence of gametocytes, since some infections might, at a given time, only contain gametocytes from one sex (or a non-optimal proportion of both sexes).

We also explored how exposure to chloroquine affects the regulation of *P. vivax* gene expression in vivo by sequencing RNA from parasites collected 8 h after the first dose of chloroquine treatment. As expected, we observed a large decrease in the proportion of reads aligned to the *P. vivax* genome when compared to samples pre-treatment. However, the magnitude of this

decrease was significantly greater than the decrease in parasitemia measured by microscopy. We hypothesize that the discrepancy between the microscopy data and the number of RNA-seq reads could be caused by counting dead or metabolically-inactive parasites, resulting in an overestimation of the parasitemia after chloroquine treatment. This hypothesis, that will need to be validated in future analyses, would suggest that clearance rates, typically determined for antimalarial drugs by microscopy, could be underestimated due to the confounding rate at which dead parasites are cleared from the circulation. In striking contrast with this important decrease in the proportion of parasite mRNAs after treatment, we noted few qualitative changes in parasite gene expression upon treatment. In fact, only a handful of genes were differentially expressed after drug treatment and the overall profiles of gene expression remained unchanged between infections. One possible explanation for this observation is that the trophozoites, who contribute the vast majority of the parasite transcripts, are unaffected by chloroquine while the other blood stages are rapidly eliminated (the decrease in the expression level of invasion genes and exported proteins after treatment could thus reflect a lower proportion of schizonts after treatment). This hypothesis would explain the similar pattern of gene expression before and after treatment, while (1) the lack of new trophozoites 8 h after treatment and (2) the continued development of trophozoites into chloroquine-susceptible schizonts would explain the observed decrease in total parasite RNAs (Fig. 4). (All *P. vivax* infections included in this study were efficiently cleared by chloroquine and we did not observe any evidence of drug resistance in these parasites[23].) This hypothesis is consistent with the results of ex vivo studies that showed that *P. vivax* trophozoites are insensitive to chloroquine[49–51] in contrast to *P. falciparum* trophozoites. This difference in the stage-susceptibility to chloroquine could indicate that the drug acts through a different mechanism in *P. vivax* than it does in *P. falciparum* (see e.g., ref. [49]). For example, we could speculate that inhibition of haemozoin synthesis by chloroquine might have a strong detrimental effect on *P. falciparum* trophozoites that have a single large food vacuole but a lesser effect on *P. vivax* trophozoites that have many small vacuoles. These findings emphasize the marked differences between *P. falciparum* and *P. vivax* and the need to specifically study *P. vivax* as the observations from one *Plasmodium* species might translate poorly to another.

Overall, our study confirmed previous analyses[21] that robust gene expression profiles could be generated directly from blood samples of vivax malaria patients and that stage composition differences between infections were unlikely to confound gene expression studies. We expanded the gene expression analysis to all patients participating in a drug efficacy study in Cambodia[22,23] and, using a statistical approach to infer the stage of origin of the transcripts, showed that parasite mRNAs overwhelmingly derived from trophozoites. Our analyses also revealed that the production of male and female gametocytes was not correlated with each other, and that the gametocyte sex ratio varied significantly among infections, providing a possible mechanism for the parasite to reduce self-fertilization. Finally and consistently with our clinical observations[23], we showed that chloroquine efficiently cleared most blood-stage parasites but had little effect on *P. vivax* trophozoites, indicating that chloroquine may act according to different mode of action in *P. vivax* than it does in *P. falciparum*. In summary, our study highlighted the biological knowledge that can be obtained from studying in vivo the gene expression profiles of *P. vivax* clinical infections and, combined with in-depth analysis of genetic diversity[22,23], showed that genomic approaches provide a promising framework to better understand this important human pathogen.

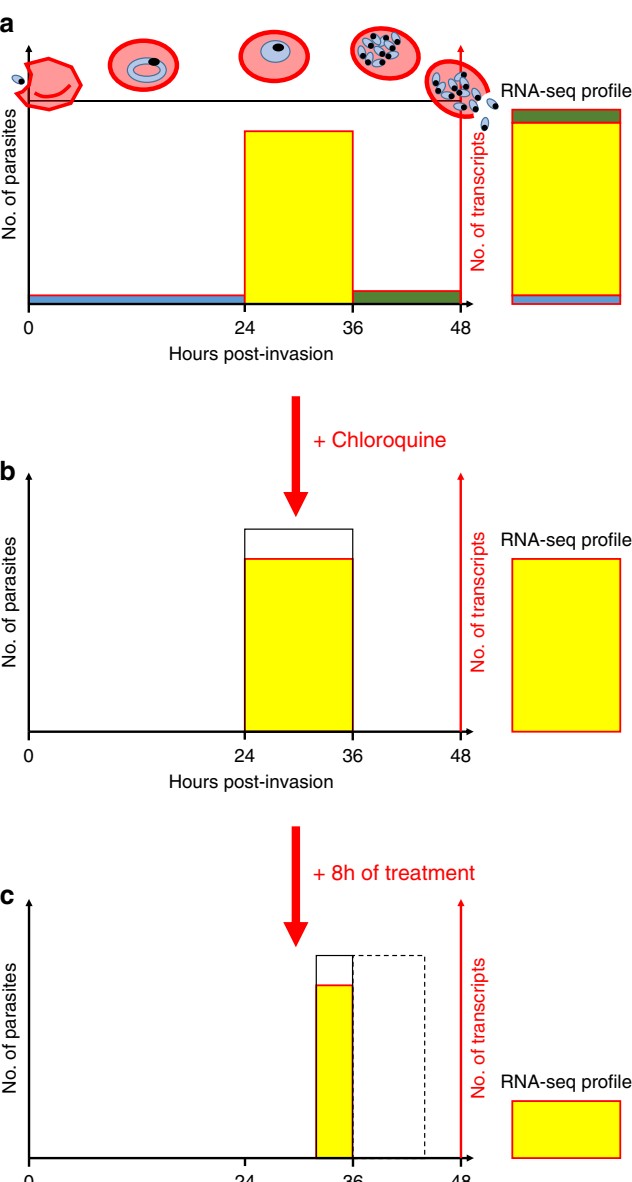

**Fig. 4** Possible effect of chloroquine on *P. vivax* blood-stage parasites. **a** The figure shows a simplified *P. vivax* infection containing uniform amount of parasites (black line) along their intraerythrocytic cycle (x-axis, in hours). The amount of mRNA produced by the different stages is represented in color (blue-rings, yellow-trophozoites, and green-schizonts) and well as the resulting RNA-seq profile that is dominated by the transcriptionally active trophozoites (right). **b** Upon treatment with chloroquine, all rings and schizonts are killed while the trophozoites are not affected. **c** Eight hours after chloroquine treatment, there are no new trophozoites, since the rings have been eliminated, and the trophozoites developing into schizonts die (dashed box), producing much fewer *P. vivax* reads by RNA-seq, but identical expression profiles

## Methods

**Patients**. We analyzed blood samples collected from 26 vivax malaria patients enrolled in a chloroquine efficacy study[23]. All patients originated from villages within 50 km of BanLung city (Ratanakiri Province, Cambodia), presented with fever (or history of fever within 48 h) and were positive for *P. vivax* DNA and no other *Plasmodium* DNA (see refs. [22,23] for details). After providing written informed consent, all patients were treated with a supervised standard 3-day course of chloroquine (30 mg/kg, Nivaquine) and monitored for 60 days. The study was

approved by the Cambodian National Ethics Committee for Health Research (038 NECHR 24/02/14) and registered at ClinicalTrials.gov (NTC02118090).

**Sample collection and stranded RNA-seq library preparation**. Upon enrollment, and prior to the first administration of chloroquine, we collected ~50 μL of blood by finger prick from all patients ($n = 26$) and immediately stored it in 500 μL of Trizol at −80 °C. Additional blood samples were collected similarly from 20 of the patients 8 h after the initial chloroquine administration.

We extracted RNA from all blood samples stored in Trizol using the Zymo Direct-zol kit with an in-column DNAse step and eluted RNA into 20 μL of water. We then prepared Illumina stranded RNA-seq libraries after ribosomal RNA and globin mRNA reduction[21] and sequenced them on a HiSeq 2500 to generate a total of 1.2 billion paired-end reads of 50 bp with a median of 24 million reads per sample (Supplementary Data 1).

**Read alignment**. We aligned all reads, first to the human genome (Hg38), then to the P01 *P. vivax* genome (PlasmoDB P01 34[24]) using Tophat2[52] with the following parameters: -g 1 (to assign each read to a single location), -I 5000 (maximum intron length), -library-type fr-firststrand (for stranded libraries), resulting in a median of 5.5 million reads per sample aligned to *P. vivax* before chloroquine and 0.5 million reads after chloroquine (Supplementary Data 1). We removed potential PCR duplicates using the samtools rmdup function to end up with a median of 3.5 million reads per sample before chloroquine and 0.27 million reads after chloroquine. For each annotated *P. vivax* gene and each sample, we determined the number of reads overlapping any exon using custom perl scripts[21]. We then transformed the raw counts into normalized transcripts per million (tpm) by dividing each gene count by the gene length (in kb) and by the sum of these values (in millions) in each sample. For further analyses, we only considered annotated *P. vivax* genes with more than 20 transcripts per million (tpm) in at least 10 samples, resulting in 4999 genes analyzed.

**Gene expression deconvolution**. To determine the proportion of mRNA molecules derived from parasites at each developmental stage in each sample, we performed gene expression deconvolution using Cibersort[26]. Briefly, by comparing the profiles of "pure" cells, Cibersort first determines the gene expression signature that characterizes each specific cell type. For our analysis, we used single-cell RNA-seq data from different stages of the *P. falciparum* intraerythrocytic developmental cycle[25]. This dataset was generated using synchronized *P. falciparum* in vitro cultures collected at different time points to represent the following stages: ring, early trophozoites, late trophozoites, and schizonts. For each stage, we selected the transcriptomes of 100 cells and normalized the data into transcripts per million before removing all genes without reads in more than 90% of the cells. We then retrieved the orthologous *P. vivax* gene names using PlasmoDB, and any genes with no orthologs or multiple orthologs were removed, resulting in a total of 2143 genes. We used this dataset to determine signature gene expression profiles of each stage (provided as Supplementary Data 4) and estimated the proportion of the different stages in different mixed samples.

Gene expression deconvolution has been extensively used to investigate cell-type heterogeneity but we verified its performance for differentiating *Plasmodium* developmental stages using several datasets. First, we analyzed RNA-seq data generated from another study of synchronized in vitro cultures of *P. falciparum* parasites[27]. Second, we generated mock bulk RNA-seq data by mixing, in different proportions, 20 parasites at different stages from a third dataset consisting of single-cell RNA-seq generated from *P. berghei*[28]. All datasets were converted to orthologous *P. vivax* genes in order to use the same signature gene set for all analyses. Finally, we used Cibersort to analyze the *P. vivax* RNA-seq profiles generated from all patient infections. For all analyses, we ran Cibersort with 100 permutations and quantile normalization disabled.

**Gene co-expression analysis**. To examine the gene expression of gametocytes, we first retrieved putative gametocyte genes from the literature and determined the Pearson correlation between the gene expression level of each pair of gametocyte genes across all samples. We then grouped these genes using unsupervised hierarchical clustering and assessed the significance of the clusters using pvclust[53]. We also identified putative novel gametocyte genes by retrieving genes whose expression was statistically correlated with one of the known gametocyte genes (with Pearson's $R > 0.8$ and $p < 0.01$).

**Determination of gene expression profile similarity**. To assess the overall effect of the first dose of chloroquine on the parasite gene expression profiles, we compared the mean difference in gene expression across all genes between paired samples, to the mean difference observed between randomly paired samples. We first calculated the pair-wise normalized gene expression differences for each pair of samples (i.e., before and after chloroquine) across all expressed genes using the following formula:

$$\sum_{\text{all genes}} \frac{X_{i,0} - X_{i,1}}{X_{i,0} + 1}$$

where $X_{i,0}$ stands for the normalized gene expression (in tpm) of the sample $i$ at time 0 for gene $X$. We then summed all pair-wise differences across all pairs of samples and compared this number to the sum of the pair-wise differences obtained by randomly pairing samples ($n = 100$) (i.e., $X_{i,0} - X_{j,1}$ where $i$ and $j$ are different patients).

**Identification of genes differentially expressed**. We also determined which *P. vivax* genes were significantly affected by chloroquine by testing for differential gene expression using EdgeR and paired analyses[54]. To account for the large difference in the number of reads obtained from *P. vivax* mRNAs before and after chloroquine treatment, we randomly subsampled all pre-treatment datasets to the same number of reads as in their corresponding post-treatment datasets. Due to the lower read counts in samples after treatment and the subsampling, only 4280 genes remained expressed above the threshold described above and were included in the analysis of the effect of chloroquine on parasite gene expression. All analyses were corrected for multiple testing using false discovery rates[55,56].

**Code availability**. All the programs, and the specific versions and parameters, used in this study are described in Methods. Custom code available at https://github.com/atomadam2/PvCQ_RNAseq.

**Reporting summary**. Further information on experimental design is available in the Nature Research Reporting Summary linked to this article.

## Data availability

The sequence data are freely available in NCBI SRA under the BioProject PRJNA378759.

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

## Acknowledgements

The authors would like to thank all patients and healthcare workers involved in this study and the staff of the Malaria Molecular Epidemiology Unit at the Institut Pasteur in Cambodia and of the National Center for Parasitology, Entomology and Malaria Control in Cambodia for their collaboration and sample collection. The authors would also like to thank Dr. Kafsack and Dr. Lawniczak for facilitating access to their scRNA-seq data. This work was funded by a National Institutes of Health—NIAID award to D.S. (R01 AI103228).

## Author contributions

A.K. performed all the molecular experiments and analyses described in this study. J.P. collected the blood samples and conducted the microscopy analyses. D.M. supervised the field study. A.K. and D.S. analyzed the data and wrote the manuscript. J.P. contributed to the manuscript preparation. The manuscript was reviewed by all authors.

## Additional information

**Competing interests:** The authors declare no competing interests.

