## [Peer Review File · Nature Communications]

Reviewers' Comments:

Reviewer #1:

Remarks to the Author:

The submitted manuscript "In vivo *P. vivax* gene expression analyses reveal stage-specific chloroquine response and differential regulation of male and female gametes" estimates parasite gene expression in directly isolated RNA from malaria infected patients. They explore the abundance of different parasite lifecycle stages by deconvoluting gene expression profiles, estimate gene clusters distinct to male and female gametocytes and explore expression changes induced by treatment with chloroquine. The paper adds exciting new tools to the profiling *vivax* gene expression. While the paper poses exciting hypothesis of their data, I would like to see more detail on some of the approaches they include.

Major Points

My main hesitation with the data presented here is with the deconvolution of gene expression. The authors extract the proportions of three developmental stages, ring, trophozoite and schizont. The analysis has been performed as if these were discrete states, however both *vivax* and *falciparum* gene expression profiles vary continuously across the 48hr cycle. It is not clear how the data from Otto et al, were binned or why they needed to be. There are also RNAseq datasets from male and female gametocytes which could have been included in the analysis. More critically there was no validation of the method. While performing this with *vivax* would be challenging, there are ample datasets from *falciparum*. I could see a simple in silico experiment where known mixtures of RNAseq profiles are generated and deconvoluted would be a powerful way to examine how accurate deconvolution is, and what reference datasets are the most appropriate. I would also suggest that "Statistical estimation of cell-cycle progression and lineage commitment in *Plasmodium falciparum* reveals a homogeneous pattern of transcription in ex vivo culture, Lemieux et al PNAS 2009" should be cited as the first example of expression profile deconvolution in patient derived malaria parasite transcriptomes.

Minor Points

Line 15: "on the biology and regulation of *Plasmodium* parasites" – its not clear what this sentence means to me, perhaps a typo?

Line 146: What was the impact of both the percentage of total *vivax* reads and the read depth on the results? Did either factor drive the PCAs?

Line 192: The decrease in reads/parasites by microscopy I found very intriguing. Firstly, could you show the correlation between the two measures in Fig. 3a, perhaps in a scatter plot? Secondly, I plain don't understand the result. Surely, if a more transcriptionally active form is enriched in the treated samples then the decrease in *vivax* reads should be less extreme than the decrease in parasitaemia, not more? If so, then are post-treatment trophozoites as transcriptionally active as their partners or have they entered some arrest in transcriptional activity?

Line 279: I also would like a greater explanation of the rationale behind the independence of male and female gametocytes. Different abundances of male and female gametocytes have been observed previously, and the skew of sex ratios has been described (i.e. Reece et al, Nature, 2008). It is innovative to be able to extract sex ratios from the data. However, I do not understand the independence claim – is this that transcriptional programs are independent? This would be assumed to be the case as both sexes are known to follow their own terminal differentiation. Or is the claim that the decision making to become a male or female gametocyte is independent of others (which does not appear to be supported by the literature or the data). Again, this is a great novel tool but would greatly benefit from more explanation.

Reviewer #2:

Remarks to the Author:

This group has pioneered genomic studies of *P. vivax* at levels of DNA and RNA and in context of population biology. I have no doubt they will lead the next crucial breakthrough applications and discoveries in this challenging and important disease organism. *P. vivax* presents massive challenges to studying it, largely due to the inability to culture it. This is particularly true for transcriptome analysis, requiring that studies are done directly from patients (*ex vivo*), creating huge difficulties in devising controlled experimental frameworks for analysis; consequently, the methods available introduce new variables that confound data interpretation. This group published the first example of this approach in a 2017 Scientific Reports paper in 3 patients and noted then the "surprising similarities of the parasite gene expression patterns across infections, despite extensive variations in parasite stage proportion".

The manuscript under review here, "In vivo *P. vivax* gene expression analyses reveal stage-specific chloroquine response and differential regulation of male and female gametocytes", is impressively done and takes the next step to a larger number of patient samples and improves the detail of their original observations; however, in terms of both novelty and definitive discovery the data as presented fall short of a significant advance that would place it at the highest level journal.

The authors make interesting observations and raise very important questions and implications from the data, however the structure of the experiments/analysis does not yield unambiguous answers. The authors refer to other submitted work pertaining to the chloroquine response that might contain details needed for robust interpretation. I do not disagree with authors' interpretation in most cases and some of the implications are indeed exciting.

For this work to have the greatest impact, several points need to be delved into rather than its current state of listing interpretations that go beyond the data. More explicitly explain that this is the state of the art, it is an important next step, the hypotheses generated here suggest key next experiments that will be big and difficult but can be done. The points below are geared to reconfigure the manuscript in that way. Importantly, there are no significant flaws to the work that was done. It is very well done and very clearly presented.

In general, the authors must do a much better job of placing this work in the context of their own work, other work in *P. vivax*, being very clear about what had been done, what is known, what is novel. And, as pointed out below, draw more heavily on the extensive data available in other malaria parasite species.

Intro:

"due to the polyclonality of most *P. vivax* infections," provide a citation.

Give a bit more basic biology detail of *P. vivax* (asynchronicity details with strong citations; does it relate to fever? Virulence? Contrast this with *P. falciparum*? Is asynchrony thought to be a 'better evolved' to the host to minimize virulence?).

Is this the 'first global perspective...in vivo'? Be more clear about these authors' earlier work (citation 11) and the Hoo et al 2016 and where this fits and what it brings.

Methods:

Provide the medians and better sense of the spread of the percentage of reads originating from *P. vivax* transcripts. What might be the significance of the fairly low correlation with parasitemia (can be it be used with visual data to consider their interpretation about stage expression more precisely)?

Explain CIBERSORT in more detail. Why is this the best option and what does the choice of this tool

rely on in terms of assumptions and, necessarily, what it is equipped to find. It is a categorical sorting method and thus is going to work in a context of broad categories. For *P. falciparum*, the definitive feature of the transcriptome is its continuous and sinusoidal nature, and Hoo et al 2016 demonstrated this for *P. vivax* in a time-course study that is not well leveraged here). This must be cited and considered carefully with respect to how the tools and their inherent assumptions will constrain the interpretation.

It may be of used to examine Adjalley et al 2015 in *P. falciparum* where they started with an analysis reselecting/expecting a cyclical nature of expression and then they stepped back to categorical classification. It would be very interesting and important if the gene expression cascade is fundamentally different in *P. vivax*, but it is arguably not the most parsimonious expectation.

The authors approach to identifying a gametocyte network is smart and given the early phase of understanding the important biology in *P. vivax*, it's a good by entry level way to recognize the presence of a signature that can be further explored in more precise ways in future studies. The methods used are largely geared to proof of concept (that gametocytes can be recognized in the transcript signals); be clear about what the next step/applications of this would be.

Results:

Fig 1A and 1B may be mis-labeled? It clearly indicates that rings are making more mRNA than trophs.

The differential gene expression analysis underscores the challenges the authors face and the conduct an impressive study be initially sampling, then treating, followed by another 8h post-treatment sample. Stage confounding will be very difficult to assess and quantitatively address in a specific way. The stage 'trophozoite' represents at least a 12h window and, for example, it could easily be that they are looking at trophs that weren't treated and comparing them to trophs that were, with their need to rely on the assumption of asynchronicity. Understandably, the authors are making sensible choices, but they need to be very clear about their assumptions and support them better and explain to the reader how important these assumptions are to the analytical approach and the interpretation of the data. This is especially true for the deconvolution algorithm.

It is crucial, but not entirely clear that their gene expression deconvolution matches precisely with the microscopy results. They state they are seeing mostly trophs by microscopy and by gene expression. This is very much a blanket statement that is necessarily ignoring rings in those samples and the potential for misleading results. It seems that the authors have a great opportunity to bounce their assumptions and observations off of the massive available *P. falciparum* data. Are they suggesting that *P. vivax* is fundamentally unique (certainly the asynchronicity is a very important difference which by itself would be a crucial biology/evolution exploration, particularly as relates to the well-adapted parasite less lethal to its host) and that even a concocted controlled study of asynchronous *P. falciparum* could not be used to calibrate an appropriate approach for *P. vivax*?

For different reasons, the role of stage in confounding identification of signal (e.g. for drug response) in *P. falciparum* transcriptome has been a significant challenge that remains unresolved. In this case, the authors are seemingly benefitting from the reverse situation, that stage effects will effectively become irrelevant in an asynchronous setting, but more likely it indicates that only the most dominant signal will be discernable.

It would be useful to align the chloroquine pre-treatment samples with the post-treatment samples in the same figure? Are the proportions the same? If there is some degree of synchronicity that could be confounding interpretation? It is noted that chloroquine does not effect troph transcription (as it does in *P. falciparum*). Consider more carefully the extent to which this is important and why/how it would be the case, vs the possibility that stage effects in the analysis between the two species could be

obscuring the comparison. How do *P. fal* ex vivo vs in vivo data compare (could also give insight to what is expected/observed in the *P. vivax*).

Is it a new finding in *P. vivax* that males and females look different by transcription? Is this different from what is seen in other malaria species? Does it bring important new biology into view and are there clear next steps for how this might be used?

Discussion:

The conclusions are relatively bold given the data and it is important for the authors to explain this, particularly to spell out the ideal data it would take to have definitive conclusions and why this work is an important step in that direction. For example, address how the troph expression profiles are predominant and in a tight enough developmental window to support the argument that this developmental stage is all one needs to collect and interpret perturbation expression profiles in *P. vivax*. Explain how this is an opportunity that is not available for *P. falciparum* for which a tightly experimentally synchronized 12h developmental stage window would be very inadequate.

Consider carefully the extent to which the deconvolution algorithm contributed to the conclusion that trophozoites contributed nearly all of the transcripts and relate it to data from other *Plasmodium* species (the 'overwhelming transcriptional signal of trophozoite parasites'). Also consider the width of the window, how many hours and what proportion of the full erythrocytic cycle.

Support the conclusion that the expression profiles are 'robust'.

Is it interesting (expected? Seen for other *Plasmodium* species) that gametocytes cluster into two groups, and that female and male gene expression is markedly different?

The implication of a mechanism to reduce the probability of self-fertilization should be considered more critically, including what the optimal data would be to reach this conclusion and how the approach here could be expanded to get these data.

It is difficult to know how meaningful the conclusions for chloroquine response are, and particularly how this information can be used. What is the take home message about chloroquine mode of action? Again, is there a possibility of contrasting this with detailed data from other malaria parasite species and addressing the extent to which it is thought that *P. vivax* is unique or would be expected to be? The best supported findings here involve the gametocyte expression, but the novelty and relevance should be discussed in more detail.

We would like to thank the editor and the reviewers for their kind assessment of our manuscript and their constructive comments that have significantly improved the study. We apologize for the delay in revising the manuscript but the new gene expression deconvolution took longer than expected. We have now addressed the reviewers' concerns in the revised version and included their suggestions. We present below a detailed answer to all their comments (the page numbers refer to the revised manuscript).

Reviewer #1 (Remarks to the Author):

The submitted manuscript "In vivo *P. vivax* gene expression analyses reveal stage-specific chloroquine response and differential regulation of male and female gametes" estimates parasite gene expression in directly isolated RNA from malaria infected patients. They explore the abundance of different parasite lifecycle stages by deconvoluting gene expression profiles, estimate gene clusters distinct to male and female gametocytes and explore expression changes induced by treatment with chloroquine. The paper adds exciting new tools to the profiling *vivax* gene expression. While the paper poses exciting hypothesis of their data, I would like to see more detail on some of the approaches they include.

Major Points

My main hesitation with the data presented here is with the deconvolution of gene expression. The authors extract the proportions of three developmental stages, ring, trophozoite and schizont. The analysis has been performed as if these were discrete states, however both *vivax* and *falciparum* gene expression profiles vary continuously across the 48hr cycle. It is not clear how the data from Otto et al, were binned or why they needed to be.

We have extensively modified the section on gene expression deconvolution to address the reviewers' comments (see also below). We had initially binned the data from Otto *et al.* since Cibersort requires at least two replicates for each profile used as reference. Based on the reviewers' comments, we have changed the reference datasets and now use instead data from single-cell RNA-seq (and the data from Otto *et al* is now used to validate the gene expression deconvolution approach).

We agree with the reviewer that gene expression varies continuously during the *Plasmodium* intraerythrocytic life cycle and that binning parasites into discrete stages is somehow artificial but this approach facilitates comparisons with microscopy data and the interpretation of the data for readers who are most familiar with this classification. We have included a detailed discussion of this limitation on line 311-318.

There are also RNAseq datasets from male and female gametocytes which could have been included in the analysis.

Given the marked differences in gametocytogenesis between *P. falciparum* and *P. vivax*, we feel a little uncomfortable in using these data to deconvolute *P. vivax* gene expression data without a better understanding on the specificities of each species. We therefore focused on the asexual stages for the deconvolution. Hopefully, scRNA-seq data from *P. vivax* will soon allow us to perform the analyses suggested.

More critically there was no validation of the method. While performing this with vivax would be challenging, there are ample datasets from falciparum. I could see a simple in silico experiment where known mixtures of RNAseq profiles are generated and deconvoluted would be a powerful way to examine how accurate deconvolution is, and what reference datasets are the most appropriate.

We thank the reviewer for this suggestion. We have now significantly edited our manuscript to include the analyses proposed (on pages 9, 10, 15 and Supplemental Information). We used *P. falciparum* single-cell RNA-seq data from the Kafsack lab (Poran *et al.*, 2018) as reference and selected gene expression data from 100 cells for each ring-, early trophozoite-, late trophozoite- and schizont datasets to perform deconvolution on the following samples:

- our *P. vivax*-infected patient samples (Figure 1A),
- the *P. falciparum* cultures studied by Otto *et al.* 2010 (Supplemental Figure 3A),
- mock mixtures of known proportions of *P. berghei* asexual parasites generated using scRNA-seq data from Reid *et al.*, 2018 (Supplemental Figure 3B).

The accuracy of the results varies between datasets and is likely influenced by many parameters (including the purity of the synchronized cultures) as discussed on page 15. But overall our analyses show that gene expression deconvolution extracts the key information from the mixture samples and works as well for differentiating *Plasmodium* parasite stages as it does for cell type heterogeneity in human tissues.

I would also suggest that “Statistical estimation of cell-cycle progression and lineage commitment in *Plasmodium falciparum* reveals a homogeneous pattern of transcription in ex vivo culture, Lemieux *et al* PNAS 2009” should be cited as the first example of expression profile deconvolution in patient derived malaria parasite transcriptomes.

We thank the reviewer for bringing this important study to our attention. We now cite this study on page 15.

Minor Points:

Line 15: “on the biology and regulation of *Plasmodium* parasites” – its not clear what this sentence means to me, perhaps a typo?

We have replaced this sentence in the abstract by “Studies of gene expression have provided invaluable insights on the biology of *Plasmodium* parasites and their response to environmental or therapeutic stimuli”

Line 146: What was the impact of both the percentage of total vivax reads and the read depth on the results? Did either factor drive the PCAs?

All principal component analyses are conducted using normalized read counts per gene (i.e., normalizing across samples for the number of reads mapped to *P. vivax* genes).

Line 192: The decrease in reads/parasites by microscopy I found very intriguing. Firstly, could you show the correlation between the two measures in Fig. 3a, perhaps in a scatter plot?

We have added this figure (new Supplemental Figure 4).

Secondly, I plain don't understand the result. Surely, if a more transcriptionally active form is enriched in the treated samples then the decrease in vivax reads should be less extreme than the decrease in parasitaemia, not more? If so, then are post-treatment trophozoites as transcriptionally active as their partners or have they entered some arrest in transcriptional activity?

At this point, we cannot say definitively what causes the discrepancy between the change in parasite read counts and the change in parasitemia measured by microscopy. We speculate that microscopy counts might include a large number of dead or transcriptionally inactive parasites (see page 17). To better explain how chloroquine affects the number of *P. vivax* RNA-seq reads but has little qualitative impact on the gene expression profile, we have now added an additional figure (Figure 4).

Line 279: I also would like a greater explanation of the rationale behind the independence of male and female gametocytes. Different abundances of male and female gametocytes have been observed previously, and the skew of sex ratios has been described (i.e. Reece et al, Nature, 2008). It is innovative to be able to extract sex ratios from the data. However, I do not understand the independence claim – is this that transcriptional programs are independent? This would be assumed to be the case as both sexes are known to follow their own terminal differentiation. Or is the claim that the decision making to become a male or female gametocyte is independent of others (which does not appear to be supported by the literature or the data). Again, this is a great novel tool but would greatly benefit from more explanation.

We thank the reviewer for this comment. We have now included the reference to Reece *et al.*, 2008 and moderated our statements on page 16 to emphasize the independence of the terminal differentiation of male and female gametocytogeneses.

Reviewer #2 (Remarks to the Author):

This group has pioneered genomic studies of *P. vivax* at levels of DNA and RNA and in context of population biology. I have no doubt they will lead the next crucial breakthrough applications and discoveries in this challenging and important disease organism. *P. vivax* presents massive challenges to studying it, largely due to the inability to culture it. This is particularly true for transcriptome analysis, requiring that studies are done directly from patients (ex vivo), creating huge difficulties is devising controlled experimental frameworks for analysis; consequently, the methods available introduce new variables that confound data interpretation. This group published the first example of this approach in a 2017 Scientific Reports paper in 3 patients and

noted then the “surprising similarities of the parasite gene expression patterns across infections, despite extensive variations in parasite stage proportion”.

The manuscript under review here, “In vivo *P. vivax* gene expression analyses reveal stage-specific chloroquine response and differential regulation of male and female gametocytes”, is impressively done and takes the next step to a larger number of patient samples and improves the detail of their original observations; however, in terms of both novelty and definitive discovery the data as presented fall short of a significant advance that would place it at the highest level journal.

The authors make interesting observations and raise very important questions and implications from the data, however the structure of the experiments/analysis does not yield unambiguous answers. The authors refer to other submitted work pertaining to the chloroquine response that might contain details needed for robust interpretation. I do not disagree with authors’ interpretation in most cases and some of the implications are indeed exciting.

Our analyses of chloroquine resistance in these same *P. vivax* infections have now been accepted for publication at JID and we have included the citation accordingly.

For this work to have the greatest impact, several points need to be delved into rather than its current state of listing interpretations that go beyond the data. More explicitly explain that this is the state of the art, it is an important next step, the hypotheses generated here suggest key next experiments that will be big and difficult but can be done. The points below are geared to reconfigure the manuscript in that way. Importantly, there are no significant flaws to the work that was done. It is very well done and very clearly presented.

In general, the authors must do a much better job of placing this work in the context of their own work, other work in *P. vivax*, being very clear about what had been done, what is known, what is novel. And, as pointed out below, draw more heavily on the extensive data available in other malaria parasite species.

Intro:

“due to the polyclonality of most *P. vivax* infections,” provide a citation.

We have now included citations for this statement on page 3.

Give a bit more basic biology detail of *P. vivax* (asynchronicity details with strong citations; does it relate to fever? Virulence? Contrast this with *P. falciparum*? Is asynchrony thought to be a ‘better evolved’ to the host to minimize virulence?).

Is this the ‘first global perspective...in vivo’? Be more clear about these authors’ earlier work (citation 11) and the Hoo et al 2016 and where this fits and what it brings.

We thank the reviewer for these important suggestions. The introduction has been edited to address these points (see page 3).

Methods:

Provide the medians and better sense of the spread of the percentage of reads originating from *P. vivax* transcripts.

The median values are now provided in the text and the complete information is available in Supplemental Table 1.

What might be the significance of the fairly low correlation with parasitemia (can be it be used with visual data to consider their interpretation about stage expression more precisely)?

We are not sure why the proportion of reads mapping to *P. vivax* is so poorly correlated with the parasitemia. Since our data suggest that the trophozoites are more transcriptionally active than the other asexual stages, we tried to correct the parasitemia for stage composition difference but this did not significantly improve the correlation. It is also possible that this poor correlation is, partially, influenced by variations in WBC counts or transcription among patients. At this stage we do not feel confident enough to speculate further on this observation.

Explain CIBERSORT in more detail. Why is this the best option and what does the choice of this tool rely on in terms of assumptions and, necessarily, what it is equipped to find. It is a categorical sorting method and thus is going to work in a context of broad categories. For *P. falciparum*, the definitive feature of the transcriptome is its continuous and sinusoidal nature, and Hoo et al 2016 demonstrated this for *P. vivax* in a time-course study that is not well leveraged here). This must be cited and considered carefully with respect to how the tools and their inherent assumptions will constrain the interpretation.

It may be of used to examine Adjalley et al 2015 in *P. falciparum* where they started with an analysis resepecting/expecting a cyclical nature of expression and then they stepped back to categorical classification. It would be very interesting and important if the gene expression cascade is fundamentally different in *P. vivax*, but it is arguably not the most parsimonious expectation.

We acknowledge the issue raised by the two reviewers regarding the continuous nature of the *Plasmodium* intraerythrocytic life cycle and the limitations of categorizing stages into discrete units. We believe however that there are some advantages in this approach, including comparisons with microscopy data as well as the possibility to soon include sexual stages (the gametocytes) in the deconvolution as more data becomes available. We now discuss this issue and cite Hoo et al. and other key references in the discussion on page 15-16. We have also significantly expanded the analyses and validation of stage deconvolution (see also our response to Reviewer 1).

The authors approach to identifying a gametocyte network is smart and given the early phase of understanding the important biology in *P. vivax*, it's a good by entry level way to recognize the

presence of a signature that can be further explored in more precise ways in future studies. The methods used are largely geared to proof of concept (that gametocytes can be recognized in the transcript signals); be clear about what the next step/applications of this would be.

We have added a section in the discussion to address this point on pages 16-17.

Results:

Fig 1A and 1B may be mis-labeled? It clearly indicates that rings are making more mRNA than trophs.

We thank the reviewer for bringing this error to our attention and apologize for this mistake. We have corrected it in the revised manuscript.

The differential gene expression analysis underscores the challenges the authors face and the conduct an impressive study by initially sampling, then treating, followed by another 8h post-treatment sample. Stage confounding will be very difficult to assess and quantitatively address in a specific way. The stage 'trophozoite' represents at least a 12h window and, for example, it could easily be that they are looking at trophs that weren't treated and comparing them to trophs that were, with their need to rely on the assumption of asynchronicity. Understandably, the authors are making sensible choices, but they need to be very clear about their assumptions and support them better and explain to the reader how important these assumptions are to the analytical approach and the interpretation of the data. This is especially true for the deconvolution algorithm.

This is a very important point. In our original analyses, we were somewhat challenged due to only having bulk RNA-seq data, and had to combine different timepoints to define specific stages, for example, trophozoites defined as 16, 24, and 32 hours after synchronization of *in vitro* culture. In the new analyses we present, we use single cell data from specific timepoints including early and late trophozoites. The result of our new analyses confirms what we showed previously: after chloroquine treatment, the stage compositions determined by RNA-seq remained largely the same. We have included this discussion on pages 9 and 11 and discuss how additional data (e.g., from scRNA-seq) will significantly improve our resolution and therefore our understanding of stage-specific responses (on page 14).

It is crucial, but not entirely clear that their gene expression deconvolution matches precisely with the microscopy results. They state they are seeing mostly trophs by microscopy and by gene expression.

This is very much a blanket statement that is necessarily ignoring rings in those samples and the potential for misleading results. It seems that the authors have a great opportunity to bounce their assumptions and observations off of the massive available *P. falciparum* data. Are they suggesting that *P. vivax* is fundamentally unique (certainly the asynchronicity is a very important difference which by itself would be a crucial biology/evolution exploration, particularly as relates to the well-adapted parasite less lethal to its host) and that even a concocted controlled study of asynchronous *P. falciparum* could not be used to calibrate an appropriate approach for *P. vivax*?

We apologize if this point was not entirely clear in the initial manuscript and we have edited the manuscript accordingly (see also discussion above on the discordance between microscopy and gene expression deconvolution).

For different reasons, the role of stage in confounding identification of signal (e.g. for drug response) in *P. falciparum* transcriptome has been a significant challenge that remains unresolved. In this case, the authors are seemingly benefitting from the reverse situation, that stage effects will effectively become irrelevant in an asynchronous setting, but more likely it indicates that only the most dominant signal will be discernable.

It would be useful to align the chloroquine pre-treatment samples with the post-treatment samples in the same figure? Are the proportions the same? If there is some degree of synchronicity that could be confounding interpretation?

Unfortunately, we do not have microscopy data from the blood samples collected 8 hours after chloroquine treatment. The stage composition inferred by gene expression deconvolution is very similar in samples before and after treatment and reflects the low susceptibility of trophozoites to chloroquine. We have also added a new figure summarizing our model for the effect of chloroquine on gene expression profiles (Figure 4).

It is noted that chloroquine does not effect troph transcription (as it does in *P. falciparum*). Consider more carefully the extent to which this is important and why/how it would be the case, vs the possibility that stage effects in the analysis between the two species could be obscuring the comparison. How do *P. fal* ex vivo vs in vivo data compare (could also give insight to what is expected/observed in the *P. vivax*).

We thank the reviewer for bringing up these suggestions/questions. While there are some controversies (related to the purity of the cultures and the duration of the exposure), the current consensus from *P. falciparum* *in vitro* experiments is that chloroquine is most efficient against *P. falciparum* trophozoites and primarily acts by inhibiting heme catabolism (which is maximal at the trophozoite stage). In *P. vivax*, only a handful of *ex vivo* studies have been conducted but they point to a different stage specificity. Our data are consistent with these results and suggest that, *in vivo* and for susceptible strains, *P. vivax* rings are susceptible but trophozoites are not (Figure 4). This could indicate that chloroquine is acting through a different mechanism in *P. vivax* than it is in *P. falciparum*, possibly through nucleic acids in the nucleus. This might not be completely unexpected given the morphological differences between *P. falciparum* and *P. vivax* trophozoites: while *P. falciparum* trophozoites usually have one single large food vacuole where all hemoglobin degradation occurs, *P. vivax* trophozoites often have many small vacuoles. Importantly for malaria control, if chloroquine does indeed act differently in *P. falciparum* and *P. vivax*, the mechanisms of resistance to this cheap and non-toxic drug would likely be different and our knowledge of *P. falciparum* chloroquine resistance would not be very useful to understand *P. vivax* chloroquine resistance. We have now included some of this discussion in the manuscript.

Is it a new finding in *P. vivax* that males and females look different by transcription? Is this different from what is seen in other malaria species? Does it bring important new biology into view and are there clear next steps for how this might be used?

The differences in gene expression between male and female gametocytes is well characterized, including by some elegant new scRNA-seq data, but our main finding is that the proportion of male and female gametocytes varies between infections, suggesting that the terminal gametocytogeneses are independently regulated. We have also included suggestions of new experiments to validate this observation on page 16-17 (e.g., by monitoring changes in the ratio of male/female gametocytes over time in one infection). We have modified and extended our discussion on this issue (see also our response to reviewer 1).

Discussion:

The conclusions are relatively bold given the data and it is important for the authors to explain this, particularly to spell out the ideal data it would take to have definitive conclusions and why this work is an important step in that direction. For example, address how the troph expression profiles are predominant and in a tight enough developmental window to support the argument that this developmental stage is all one needs to collect and interpret perturbation expression profiles in *P. vivax*. Explain how this is an opportunity that is not available for *P. falciparum* for which a tightly experimentally synchronized 12h developmental stage window would be very inadequate.

Consider carefully the extent to which the deconvolution algorithm contributed to the conclusion that trophozoites contributed nearly all of the transcripts and relate it to data from other Plasmodium species (the 'overwhelming transcriptional signal of trophozoite parasites'). Also consider the width of the window, how many hours and what proportion of the full erythrocytic cycle.

We have edited these sections considerably to include new analyses using single-cell RNA-seq data.

Support the conclusion that the expression profiles are 'robust'.

We have removed this term.

Is it interesting (expected? Seen for other Plasmodium species) that gametocytes cluster into two groups, and that female and male gene expression is markedly different?

The implication of a mechanism to reduce the probability of self-fertilization should be considered more critically, including what the optimal data would be to reach this conclusion and how the approach here could be expanded to get these data.

The differences between female and male gametocyte gene expression is not unexpected but what is more striking is that, thanks to these differences, we identified highly variable proportions of male/female gametocytes across infections, suggesting that they are not

produced evenly. We have now clarified this in the discussion and present possible experiments to test the self-fertilization hypothesis (e.g., following one infection over time).

It is difficult to know how meaningful the conclusions for chloroquine response are, and particularly how this information can be used. What is the take home message about chloroquine mode of action? Again, is there a possibility of contrasting this with detailed data from other malaria parasite species and addressing the extent to which it is thought that *P. vivax* is unique or would be expected to be?

The best supported findings here involve the gametocyte expression, but the novelty and relevance should be discussed in more detail.

We have edited the paper to clarify our conclusions. Our data show that chloroquine very rapidly clears *P. vivax* parasites *in vivo* (with a dramatic reduction within 8 hours of the first dose), and corroborate *in vitro* data suggesting that *P. vivax* trophozoites are little affected by chloroquine (though they are also cleared from the circulation as they differentiate into schizonts). We have now included a new figure in the manuscript to emphasize these findings.

Reviewers' Comments:

Reviewer #1:

Remarks to the Author:

The authors add an impressive amount of additional analysis to this work to address comments during the initial round of reviews. The authors should check that the labels of supplementary figures matches the text (I noted SFig 2 was labelled twice in my PDF). The revised version adds greater clarity and robustness in understanding the ciphersort method. The revised paper is an exciting advance and I hope to see it in press shortly

Reviewer #2:

Remarks to the Author:

The authors conducted significant reanalysis pertaining to stage deconvolution and use some analysis of independent datasets to validate; this approach addresses that they can effectively capture bins of transcripts, i.e. tag them broadly to developmental stage. This is an important improvement; and does support that asynchronicity might not be an overwhelming terrible confounder, but it says little about how the importance of other stages is accounted for, and more importantly whether the progression through trophozoites can be more finely parsed.

In general, the authors' additions and modifications are effective in outlining the challenges and limitations of conducting these studies on a pathogen that must be studied directly from the patient (particularly their expanded discussion of the deconvolution data is well done). However, these same important limitations remain (have not changed significantly from the original submission), but have not been thoroughly addressed. Outlined below, the authors must be more clear about the important biological inferences drawn from the data are dependent on the technical limitations of the data and the analysis. These main biological inferences are: that stage (intraerythrocytic development) divergences aren't relevant for ex vivo *P. vivax* work because most transcripts are from trophozoites, and also concerning chloroquine's impact on transcription.

Notably, because the rebuttal did not include the specific edits, but rather a general statement and a list of lines to refer to in the manuscript, it is difficult to evaluate the details and extent of the improvements. This required manually aligning the pre- and post- documents; consequently, in some cases the rebuttal points are not compellingly argued or clear.

Specific comments:

Explain exactly the strengths and limitations of the deconvolution validations per se (what is being validated and how does it strengthen the biological interpretations in this manuscript?); also, explain more precisely the issue of stage resolution on the biological inference, especially the biology that will be necessarily missed. Finally, consider specifically the role of sensitivity (i.e. information that is swamped out by trophozoites) and what that could mean for understanding biology.

Can the authors parse (sub-bin) the trophozoite stages? This would be very helpful. Given they are arguing that synchronicity will not be confounding because the vast majority of mRNA is from trophs, it would be valuable to show that these can be parsed more finely. This must be DIRECTLY considered beyond the attempt at validating the deconvolution...it is not surprising (or interesting) that it is possible to bin the data; however, if they are blind to stage, even the wide variation in trophozoite development, what would be the impact on their overall interpretation about different infections (stage composition)?

The binning using single cell is an improvement. They need to support that the identified bins (directed deconvolutions) are functionally identical (or distinct) in terms of expression profiles and whether the synchronicity of expression profiles recovered is solid for making biological inferences with confidence (that stage is not confounding).

The authors still have not provided a solid explanation for the incongruence between microscopy data and transcription data. These were clearly asked for by both reviewers. Consider more carefully how this limitation could be impacting the interpretation of the chloroquine perturbation and its biological relevance (how this information can be used, built upon). Their method says schizonts are present when they're not present by microscopy (Fig 1A and 1B). This can be accounted for by lack of sensitivity? Also a mention of lack of early trophs is required.

Male and female gametocytogenesis is independently regulated work is interesting and well done. The authors do not make a strong case for the novelty and definitive discovery the data as representing significant advances (of their own work!). They should DIRECTLY state what distinctions and advances are present here vs their 2017 Science Reports work and their 2 very recent chloroquine studies.

Reviewer #2 (Remarks to the Author):

Specific comments:

Explain exactly the strengths and limitations of the deconvolution validations per se (what is being validated and how does it strengthen the biological interpretations in this manuscript?);

The analyses presented on Supplemental Figure 3 confirmed that gene expression deconvolution allows inferring the origin of the transcripts characterized by bulk RNA-seq from a mixture of parasites in different developmental stages. As indicated on line 122-124, these analyses validate the approach used and therefore the conclusions drawn from the *in vivo* gene expression deconvolution. Biologically, these analyses revealed/confirmed that all *P. vivax* blood stages are not equally transcriptionally active and that the vast majority of transcripts in asynchronous infections derive from trophozoites (suggesting that the asynchronous nature and heterogeneity of *P. vivax* infections are unlikely to massively confound *in vivo* gene expression analyses). In addition, the gene expression deconvolution analyses were critical for us to reconcile the apparent discrepancy between the remarkable decrease in *P. vivax* reads sequenced after CQ treatment and the lack of qualitative changes in gene expression.

The deconvolution validations have however some clear limitations, due primarily to the “reference datasets” available and, as the reviewer noted, the binning into discrete and crude developmental categories. While we believe that this approach represents a significant improvement for analyzing malaria parasites gene expression *in vivo*, we fully acknowledge that the approach, and our conclusions, will need to be refined in the future as better reference datasets, from more finely defined stages, become available (see also below).

This discussion is now expanded and highlighted in pages 10 and 14-16. In addition, we now provide the contribution of each gene to the stage signature in Supplementary Data files and reference them in the Methods section on lines 119-120 as well as in line 324 as requested by the editor.

Also, explain more precisely the issue of stage resolution on the biological inference, especially the biology that will be necessarily missed.

The reviewer is correct that the limited stage resolution, and the use of crudely-defined developmental stages, may fail to fully reveal the detailed mechanisms in place. For example, our analyses of the consequence of CQ on parasite transcription suggest that broadly-defined “trophozoites” are not transcriptionally affected by CQ. However, we are not able, with the current dataset available, to identify possible differences in parasite response across this extended developmental stage. This is now discussed and highlighted on page 16.

Finally, consider specifically the role of sensitivity (i.e. information that is swamped out by trophozoites) and what that could mean for understanding biology.

The difference in transcriptional activity between stages will likely complicate investigating changes in gene expression occurring in ring-stage parasites or schizonts since their signals will be swamped by

that of the trophozoites (though our analysis of gametocyte genes shows that this is not be impossible). We discussed and highlighted these issues in more extensive details on page 15.

Can the authors parse (sub-bin) the trophozoite stages? This would be very helpful. Given they are arguing that synchronicity will not be confounding because the vast majority of mRNA is from trophs, it would be valuable to show that these can be parsed more finely. This must be DIRECTLY considered beyond the attempt at validating the deconvolution...it is not surprising (or interesting) that it is possible to bin the data; however, if they are blind to stage, even the wide variation in trophozoite development, what would be the impact on their overall interpretation about different infections (stage composition)?

The analyses suggested by the reviewer would indeed be very interesting to conduct but are difficult for us to implement as they cannot be performed with the data we generated (*in vivo P. vivax* RNA-seq) but would rely on re-analysis of published data. We used scRNA-seq from published datasets to “bin” and validate the developmental stages but restricted our analyses to the categories used by the authors of the respective studies (including two “bins” of trophozoites). Since these groups are actively pursuing these questions and are working on generating a finer characterization of the gene expression changes during the intra-erythrocytic cycle, we would prefer to let these investigators complete their studies and concentrate the analyses of the present manuscript on our own *P. vivax* data.

The binning using single cell is an improvement. They need to support that the identified bins (directed deconvolutions) are functionally identical (or distinct) in terms of expression profiles and whether the synchronicity of expression profiles recovered is solid for making biological inferences with confidence (that stage is not confounding).

We expect that the bins still encompass a heterogeneous set of parasites with different expression profiles (though the bulk of the transcriptional heterogeneity is likely/probably captured by the crude binning) and, as the reviewer suggested in her/his last point, further binning will be required to fully rule out stage confounding. However, we do not believe that these points alter the main conclusions of our study, though these could be further refined with better scRNA-seq data. We have further expanded the discussion on this limitation on page 16.

The authors still have not provided a solid explanation for the incongruence between microscopy data and transcription data. These were clearly asked for by both reviewers. Consider more carefully how this limitation could be impacting the interpretation of the chloroquine perturbation and its biological relevance (how this information can be used, built upon). Their method says schizonts are present when they're not present by microscopy (Fig 1A and 1B). This can be accounted for by lack of sensitivity? Also a mention of lack of early trophs is required.

The incongruence between microscopy data and transcription data is probably caused primarily by the fact that each stage is not equally transcriptionally-active and, therefore, the proportion of stages observed by microscopy does not reflect the proportion of the transcripts originating to each stage. In addition, this discrepancy could also be caused by the low resolution of microscopy due to the low parasitemia of *P. vivax* infections: with 200 fields analyzed, microscopy only provide information based on ~50 observed parasites (for a parasitemia of 0.1%). Microscopy is therefore likely to miss or misestimate lowly abundant stages such as the schizonts on Figure 1 (using 50 uL of blood we are “sampling” 250,000 parasites by RNA-seq vs. 50 by microscopy). This discussion has been expanded and is highlighted on pages 14-15.

Male and female gametocytogenesis is independently regulated work is interesting and well done. The authors do not make a strong case for the novelty and definitive discovery the data as representing significant advances (of their own work!). They should DIRECTLY state what distinctions and advances are present here vs their 2017 Science Reports work and their 2 very recent chloroquine studies.

We have further expanded the discussion to emphasize the conclusion of the gametocyte analysis on page 18. In addition, we have modified the introduction (page 4) and conclusion (pages 19-20) of the manuscript to put this study into better context compared to our recent work.